# Intracellular Ca^2+^-Mediated AE2 Is Involved in the Vectorial Movement of HaCaT Keratinocyte

**DOI:** 10.3390/ijms21228429

**Published:** 2020-11-10

**Authors:** Soyoung Hwang, Dong Min Shin, Jeong Hee Hong

**Affiliations:** 1Department of Physiology, College of Medicine, Gachon University, Incheon 21999, Korea; snrntlwy1004@gmail.com; 2Department of Oral Biology, Yonsei University College of Dentistry, Seoul 03722, Korea; 3Department of Health Sciences and Technology, GAIHST, Gachon University, 155 Getbeolro, Yeonsu-gu, Incheon 21999, Korea

**Keywords:** keratinocyte migration, anion exchanger 2, histamine, intracellular Ca^2+^

## Abstract

Keratinocyte migration is initiated toward the wound skin barrier as a crucial process in wound healing. However, the migratory machinery used by keratinocytes is relatively unknown. Histamine signaling, including an increase in the Ca^2+^ signal, mediated the enhanced protein expression and chloride/bicarbonate exchange activity of anion exchanger AE2 in keratinocytes. In this study, we applied an agarose spot assay to induce a vectorial motion. The vectorial stimulation of the histamine-containing agarose spot enhanced the HaCaT keratinocyte migration, compared to non-directional stimulation. AE2 is associated with the vectorial movement of HaCaT keratinocytes. Enhanced expression of AE2 was mainly associated with an increase in Ca^2+^ and was abolished by the treatment with the Ca^2+^ chelating agent BAPTA-AM. These findings revealed that the directionality of Ca^2+^-exerted stimulation can play a prominent role in facilitating migration through the involvement of AE2 as a migratory machinery in HaCaT keratinocytes.

## 1. Introduction

The skin barrier is the first defense barrier that protects the body from exogenous stimuli. When the skin barrier is disturbed, keratinocyte migration is immediately initiated as a crucial protection process [1]. The migratory function of keratinocytes is modulated by the stimulation of inflammatory mediators from immune cells, such as mast cells [2]. Among the inflammatory mediators, histamine exerts its effects by activating G protein-coupled receptors, H1 to H4 receptors (H1R–H4R), which induce the release of Ca^2+^ and subsequent inflammatory reactions, including the release of several cytokines in keratinocytes [3].

Anion exchangers (AEs) are involved in pH regulation through the exchange of chloride and bicarbonate [4]. AEs are encoded by the SLC4A family, AE1–4, and are identified in various tissues. AE2 is broadly expressed in various cell types. The expression of AE2 is involved in the maintenance of the intracellular pH through the exchange of bicarbonate ions [5,6]. In addition to the classical role of bicarbonate transporters in the cellular pH maintenance, cell volume regulation is a dominant function of these transporters for cellular migration [7]. The modulation of the cellular pH is a basal function for the maintenance of cell fate, as well as for migration. Inflammation occurs when the surrounding matrix environment is acidic and hypoxic [8]. In addition, an increase in intracellular Ca^2+^ mediates cellular migration [9]. The precise role of inflammatory mediators or of the involvement of their subsequent inflammatory Ca^2+^ signaling and transporters in migratory machinery still need to be determined. Various reports have addressed the bio-mechanistic regulation of keratinocyte migration [10,11,12,13,14,15]. Unfortunately, the expression or function of AE in keratinocyte migration has not been elucidated from a migratory machinery perspective. Cellular migration or invasion is mediated by migratory machinery through the involvement of ion transporters and exchangers [16]. Therefore, we hypothesized that involvement of bicarbonate transporters is crucial for regulating the migratory machinery of keratinocytes.

The aim of the current study was to assess the role of the bicarbonate transporter in keratinocyte migration. We assessed whether increased Ca^2+^ signals via histamine stimulation enhanced bicarbonate transporter AE2 as migratory machinery mainly expressed in HaCaT keratinocytes, and whether vectorial or directional stimulation provides a migration potential to keratinocytes, in response to a wound.

## 2. Results

### 2.1. AE2 Is Activated by Stimulation of Histamine in Keratinocytes

The bicarbonate transporter in keratinocytes has rarely been examined. Thus, we first investigated the expression and role of bicarbonate transporters of SLC4 family in keratinocytes. HaCaT keratinocytes were dominantly expressed in AE2 and AE3, encoded by *SLC4A2* and *SLC4A3*, not AE1 (*SLC4A1*) and AE4 (*SLC4A9*) (Figure 1A). AE2 mRNA and protein were enhanced by the stimulation of histamine (Figure 1B,C,E). Expression of AE3 was no statistical difference in histamine stimulation (Figure 1B,D,E). It has been reported that AE3 is found in excitable tissues such as heart, brain, and smooth muscle [17]. Thus, we focused on the AE2. To confirm this, we used Metamorph software to systematically measure the intensity of AE2 with rhodamine fluorescence. The immunostaining of AE2 was also enhanced in histamine-stimulated HaCaT cells (Figure 1F,G). Although we expected the membranous expression of AE2, expression of AE2 was cytosolic in this experimental condition. It has been reported that AE2 is highly immunodetected in the basal and first suprabasal keratinocyte layer of the human epidermis (Protein Atlas website data). The chloride bicarbonate exchanger (CBE) activity was driven by the SLC4A family. Thus, the activity of AE2 was determined based on the CBE activity, as previously described [18]. CBE activity was enhanced by the stimulation of histamine, whereas it was inhibited by the treatment of Ca^2+^ chelating agent BAPTA-AM in HaCaT cells (Figure 1H,I). Enhanced CBE activity by the treatment of histamine was also confirmed in primary keratinocytes (Figure 1J,K). The primary keratinocytes were expressed AE2 protein and slightly enhanced expression by the treatment of histamine; however, AE3 protein was not observed (Figure 1L,M). Thus, role of AE2 was considered in following experiments. These results suggest that histamine stimulation enhanced AE2 expression and CBE activity in keratinocytes.

### 2.2. Increased Calcium Levels Induced by Histamine Stimulation Enhanced CBE Activity and AE2 Expression in HaCaT Keratinocytes 

Histamine exerts its effects by activating G protein-coupled receptor (GPCR), H1 to H4 receptor (H1R-H4R), which induces Ca^2+^ release and, ultimately, initiates inflammatory processes [19]. To determine the histamine-mediated Ca^2+^ signaling, the mRNA expression of histamine receptor was examined in HaCaT keratinocytes. HaCaT keratinocytes expressed the mRNAs of H1R, H2R, and H3R (Appendix A). It has been well established that histamine stimulation is primarily associated with H1R in primary keratinocytes [20]. HaCaT keratinocytes were stimulated with histamine for 24 and 48 h. The increased Ca^2+^ signals showed oscillatory spikes in the application of histamine (Figure 2A,B). The basal Ca^2+^ level increased in a time-dependent manner (Figure 2C). To confirm the Ca^2+^-dependent CBE activity, keratinocytes were stimulated with 3 mM CaCl_2_ for 48 h in presence or absence of BAPTA-AM. The CBE activity was enhanced by the treatment with 3 mM CaCl_2_, whereas reduced by the cotreatment of BAPTA-AM (Figure 2D,E). AE2 protein expression was also increased by using 3 mM CaCl_2_ (Figure 2F,G). These results indicated that the enhanced Ca^2+^ concentration increased AE2 expression and its CBE activity. 

### 2.3. CaCl_2_ and Histamine Stimulation Enhanced HaCaT Cell Migration

To examine the migration of HaCaT cells in presence of histamine and CaCl_2_ stimulation, agarose spots assay was launched. Some migration occurred in the control group and was due to the motile characteristics of keratinocytes. HaCaT keratinocyte migration was enhanced in histamine treatment, whereas attenuated by the treatment of BAPTA-AM in media (Figure 3A,B). HaCaT keratinocyte migration toward agarose was also enhanced by the treatment of CaCl_2_ in media (Figure 3C,D). The CaCl_2_-mediated migration was attenuated by the application of BAPTA-AM in media (Figure 3A–D).

### 2.4. AE2 Expression Was Dependent on Histamine and CaCl_2_ Stimulation

Histamine signaling and enhanced Ca^2+^ concentration induced cellular migration. Cellular migration is mediated by various transporters. To assess this, we evaluated the role of highly expressed AE2 on HaCaT migration. HaCaT cells were stained with AE2 antibody. The AE2 expression was enhanced by the treatment of histamine and CaCl_2_ and the co-treatment of BAPTA-AM attenuated the AE2 expression in HaCaT cells (Figure 4A,B). These results addressed that AE2 expression was related to enhanced migratory ability.

### 2.5. Motility of HaCaT Cells Was Driven More by Vectorial Chemotaxis

Despite the motile features of keratinocytes by stimulation of GPCR and Ca^2+^ signaling, migration can be directed toward a chemotactic gradient, as that used by neutrophils [7]. This prompted us to investigate the relationship between migration and gradient of stimulation, in particular. To evaluate the ability of motion of directional stimulation, called vectorial chemotaxis, in HaCaT keratinocytes, histamine stimulation was compared between non-vectorial (stimulation in media) and vectorial stimulation (histamine-containing agarose) (Figure 5A–D). Chemotactic gradient by histamine-including agarose spots enhanced the motility of HaCaT keratinocytes, compared to a non-vectorial stimulation (Figure 5D). The concept of vectorial stimulation is illustrated in Figure 5E,F. Direct stimulation of histamine provided the ability to migrate, whereas potent and facilitated migration of HaCaT keratinocytes occurred during vectorial stimulation toward histamine-containing agarose.

### 2.6. Inhibition of Transporters by DIDS Attenuated HaCaT Cell Migration

We next explored how cellular migration might occur via motile machinery. To confirm the role of transporters in migration, an anion exchanger blocker DIDS was used. The vectorial migration of HaCaT cells toward the histamine-containing agarose spot was abolished in the presence of DIDS (Figure 6A). CBE activity was also reduced in the presence of DIDS compared to the control (Figure 6B,C).

### 2.7. AE2 Inhibition Attenuated the Vectorial Movement of HaCaT Cells

We evaluated the role of AE2 on vectorial HaCaT migration. The effective inhibitor of AE2 was not available. Accordingly, we proposed and previously reported that potential anticancer agent disulfiram (DSF) attenuated the AE2 expression in lung and breast cancer cells [21]. The treatment of DSF reduced AE2 expression in HaCaT cells (Figure 7A,B). The cell viability was not affected by the treatment of DSF (Figure 7C). The HaCaT migration was also attenuated in presence of DSF toward histamine-containing agarose spot (Figure 7D,E). These findings addressed that the vectorial movement of HaCaT toward histamine-containing agarose spots is modulated by the involvement of AE2.

### 2.8. Overexpressed AE2 Enhanced Vectorial HaCaT Migration

To confirm the role of AE2 on vectorial migration, HaCaT cells were transfected AE2. The overexpressed AE2 enhanced cellular migration toward histamine-containing agarose (Figure 8A,B). Protein and mRNA expression of AE2 in AE2-overexpressed HaCaT cells were confirmed (Figure 8C,D). These data addressed enhanced AE2 provided the facilitated vectorial movement in HaCaT cells.

## 3. Discussion

The ubiquitously expressed AE2 has a fundamental role in maintaining intracellular pH levels. In this study, the increase of Ca^2+^ by inflammatory mediators, such as histamine, enhanced AE2 expression, and activity in keratinocytes. Although both AE2 and AE3 were expressed in HaCaT cells, AE2 was statistically enhanced by the histamine stimulation. We have shown that, for the first vectorial stimulation of histamine-containing agarose spot, the keratinocyte migration through the involvement of AE2 is enhanced. The enhanced expression of AE2 is mainly associated with an increase in Ca^2+^. The directionality of Ca^2+^-induced stimulation can play a prominent role in facilitated migration through the involvement of AE2 and its enhanced CBE activity. Our experimental observation was represented to the schematic illustration (Figure 9).

More recently, there have been many efforts to verify the molecular pathway of keratinocyte migration. Although few studies were addressed, Keratin 6 regulates the collective keratinocyte migration [10], nitric oxide-mediated cGMP-PKG signaling [15], JMJD3/NF-ΚB-activated Notch pathway [14], BNIP3 modulation in hypoxia-induced keratinocyte migration [13], and the negative regulation of transmembrane protein CD9 in keratinocyte migration [11]. The regulation of these molecular pathways induces cellular migration driven by the modulation of migratory machinery. The association of AE2 in migratory machinery provides a more diverse approach to regulation of migration. Of course, not only the contribution of AE2, but also the involvement of other collaborations—including sodium/hydrogen exchanger NHE, sodium/potassium/chloride cotransporter NKCC, and sodium/bicarbonate cotransporter NBC—are related to cellular migration [7]. Information on the molecular mechanism of keratinocytes within the scope of transporters is relatively rare. In this study, we demonstrated that the involvement of AE2 plays a crucial role in migratory dynamics. Besides its migratory role, AE2 can be a candidate for a challengeable transporter in keratinocytes, as an acid loader. NHE1 has been reported to be enhanced in wounded stratum corneum and to regulate the stratum corneum pH gradient [22,23]. The fundamental role of NHE1 is to maintain a steady-state intracellular pH, and this regulation can be mediated by the additional involvement of other transporters, such as NBCn1 [24]. It is necessary to evaluate the effect of supportive transporters including AE2 on pH regulation in keratinocytes in future studies.

In summary, an enhanced expression and activity of AE2 in a histamine or Ca^2+^-stimulated in vitro study of HaCaT keratinocytes may provide various insights into the pathophysiological regulation of transporters. Keratinocytes at wound edge tends to migrate and increase of speed of migration [10]. Our data addressed in this study provide the strategy for enhancement of migration. The vectorial stimulation of histamine may provide a preferred directionality of movement, and keratinocytes may utilize Ca^2+^-dependent AE2 machinery. The enhancement of AE2 may provide the efficient cellular migration and subsequent wound closure.

## 4. Materials and Methods

### 4.1. Reagents and Plasmids

β-actin antibody (A5441) and 4,4′-Diisothiocyano-2,2′-stilbenedisulfonic acid (DIDS, 462268), histamine (H7125) and Disulfiram (DSF, European Pharmacopoeia (EP) Reference Standard, D2950000) were purchased from Sigma Aldrich (Saint-Louis, MO, USA). AE2 antibody (ab42687) and AE3 antibody (ab187102) were purchased from Abcam (Cambridge, MA, USA). Calcium chloride (CaCl_2_) dihydrate was purchased from Amresco (West Chester, PA, USA). The pH indicator 2′,7′-bis-(carboxyethyl)-5-(and-6)-carboxyfluorescein (BCECF)-AM and the Ca^2+^ indicator Fura2-AM were purchased from TEFlabs (Austin, TX, USA). Pluronic acid (F-127, 20% in dimethyl sulfoxide, P-3000MP) and 1,2-bis (2-aminophenoxy) ethane-N,N,N′,N′-tetraacetic acid tetrakis, acetoxymethyl ester (BAPTA-AM) were purchased from Invitrogen (Carlsbad, CA, USA). All other used chemicals that were not mentioned here were purchased from Sigma Aldrich. Rat AE2 (HA-pcDNA3.1(+)) clone was provided by the Dr. Shmuel Muallem at National Institutes of Health (Bethesda, MD, USA) and possess 94% homologous sequence compared to human AE2.

### 4.2. HaCaT and Primary Keratinocytes Culture

The human keratinocyte cell line HaCaT was maintained in Dulbecco’s modified Eagle’s medium (DMEM, 11995073, Thermo Fisher Scientific, Waltham, MA, USA), containing 10% FBS (1600-044, Invitrogen, Waltham, MA, USA) and 100 U/mL penicillin-streptomycin (15140122, Invitrogen), and was incubated at 37 °C in a humidified incubator with 5% CO_2_ and 95% air. When the keratinocytes reached an 80% confluence, the culture medium was removed, and the keratinocytes were washed with Dulbecco’s phosphate-buffered saline (DPBS, LB001-02, Welgene, Gyeongsan-Si, Korea), followed by treatment with trypsin/ethylenediaminetetraacetic acid (EDTA) for 5 min. The dispersed keratinocytes were transferred to new culture dishes for Western blotting, agarose spot assay, and fluorescent imaging.

The human primary epidermal keratinocyte cell line-adult, HEKa (PCS-200-011, ATCC, Manassas, VA, USA) was maintained in dermal cell basal medium (PCS-200-030, ATCC, Manassas, VA, USA), containing keratinocyte growth kit (PCS-200-040, ATCC, Manassas, VA, USA) and penicillin-streptomycin-amphotericin B solution (PCS-999-002, ATCC, Manassas, VA, USA), and was incubated at 37 °C in a humidified incubator with 5% CO_2_ and 95% air. When the primary keratinocytes reached an 80% confluence, the culture medium was removed, and the keratinocytes were washed with Dulbecco’s phosphate-buffered saline (DPBS, LB001-02, Welgene, Gyeongsan-Si, Korea), followed by treatment with trypsin/ethylenediaminetetraacetic acid (EDTA) for primary cells (PCS-999-003) for 6 min. When cells appear to have detached, add the trypsin neutralizing solution (ATCC PCS-999-004) to flask. The dispersed primary keratinocytes were transferred to new culture dishes for qRT-PCR.

### 4.3. Agarose Spot Assay for Cell Migration

Cell migration was examined by performing an agarose spot assay, as described previously, by modifying the protocol of the chemotactic invasion assay. Briefly, 10 mg of agarose (UltraKem LE, Young Sciences, Korea) were placed into a 50 mL conical tube and diluted in 2 mL of physiological salt solution (PSS; 140 mM sodium chloride [NaCl], 10 mM glucose, 5 mM potassium chloride KCl, 1 mM magnesium chloride MgCl_2_, 1 mM calcium chloride CaCl_2_, 10 mM HEPES, pH 7.4, 310 mOsm) to prepare a 0.5% agarose solution, which was spotted (four spots per plate) onto six-well plates (Thermo Fisher Scientific, Waltham, MA, USA) and allowed to cool down for 8 min at 4 °C. Keratinocytes (4 × 10^5^) were then plated and allowed to adhere for 4 h before adding DMEM containing 0.1% FBS (Invitrogen, Waltham, MA, USA) and 100 U/mL penicillin (Invitrogen, Waltham, MA, USA). After 4, 24, 48, and 72 h at 37 °C, images were collected using the Meta Morph software (Molecular Devices) with a 10 × objective (Olympus, Tokyo, Japan). Keratinocytes that appeared underneath the agarose spot were counted as keratinocytes that had migrated.

### 4.4. Measurement of the CBE Activity of AE2

Keratinocytes were attached onto coverslips and loaded onto the chamber with 6 μM BCECF-AM (TEFlabs, Austin, TX, USA) in the presence of 0.05% pluronic acid (Invitrogen, Waltham, MA, USA) for 15 min at room temperature. After stabilization of the fluorescence, the keratinocytes were subjected to perfusion using solution A for, at least, 5 min prior to intracellular pH (pH_i_) measurements. pH_i_ was measured based on the BCECF fluorescence, using dual excitation wavelengths (495 and 440 nm) and an emission wavelength (530 nm). Chloride bicarbonate exchanger (CBE) activity of AE2 was determined using a Cl^-^-free HCO_3_^-^-buffered solution containing 126 mM Na^+^. The keratinocytes were incubated with a CO_2_-saturated HCO_3_^-^-buffered solution for the acidification of the cytosol, and then perfused with a Cl^-^-free HCO_3_^-^-buffered solution. The pH measurement-based CBE activity was calculated from the slope of the pH_i_ increase during the first 30–45 s in Cl^−^-free HCO_3_^−^-buffered solution, and was expressed as the percent fold change relative to that of the CBE activity of the control, as described previously [18]. Images were obtained at an interval of 1 s using a high resolution-CCD camera (Retiga 6000, Q-Imaging, Surrey, BC, Canada) linked to an inverted microscope (Olympus, Tokyo, Japan) and analyzed using a MetaFluor system (Molecular Devices, Downingtown, PA, USA). Each image was normalized by subtracting the background fluorescence from the raw background signals.

### 4.5. Measurement of Intracellular Calcium Increase

HaCaT cells (4 × 10^5^) were seeded onto coverslips and treated with 4 μM Fura-2/AM in the presence of 0.05% Pluronic F-127 for 15 min in PSS at room temperature, in the dark. Changes in [Ca^2+^]_i_ were determined by measuring the fluorescence intensities using dual excitation wavelengths (340 and 380 nm) and an emission wavelength (510 nm). The results are presented as fluorescence (F) ratios (ratio = 340/380). The emitted fluorescence was monitored using a CCD camera (Retiga 6000, Q-Imaging, Surrey, BC, Canada) attached to an inverted microscope (Olympus, Tokyo, Japan) and analyzed using a MetaFluor system (Molecular Devices, Downingtown, PA, USA). Fluorescence images were obtained at 1 s intervals, and the background fluorescence at each excitation wavelength was subtracted from the raw signals. Evoked [Ca^2+^]_i_ (ΔCa^2+^) responses were calculated by dividing the maximum Ca^2+^ peak of the agonist in the presence of histamine by the maximum Ca^2+^ peak of agonist stimulation.

### 4.6. Western Blotting

The keratinocytes, which were treated with histamine, DSF, or DIDS (Sigma Aldrich, Saint-Louis, MO, USA), were incubated with 1 × lysis buffer (Cell signaling) containing 20 mM Tris, 150 mM NaCl, 2 mM EDTA, 1% Triton X-100, and a protease inhibitor mixture for 5 min at room temperature. The keratinocytes were sonicated and centrifuged at 11,000× *g* for 15 min at 4 °C, and protein concentration was determined by using the Bradford assay (Bio-Rad, Hercules, CA, USA). The lysed samples were incubated with protein sample buffer at 37 °C for 15 min. The warmed protein samples (30 μg) were subjected to separation using sodium dodecyl sulfate-polyacrylamide gel electrophoresis (SDS-PAGE) and then transferred onto polyvinylidene difluoride (PVDF, 1620177, Bio-Rad) membranes soaked in methanol. The membrane was blocked with a 5% nonfat milk solution in Tris-buffered saline (TBS) and 0.5% Tween-20 (TBS-T) for 1 h. Then, the membrane was incubated with β-actin, AE2, and AE3 antibodies overnight at 4 °C and washed thrice with TBS-T. Following the washes, the membranes were incubated with horseradish peroxidase (HRP)-conjugated anti-mouse and anti-rabbit secondary antibodies and the protein bands were visualized using the enhanced luminescence solution (32209, Thermo Fisher Scientific, Waltham, MA, USA).

### 4.7. Confocal Imaging

Transfected HaCaT cells were transferred onto cover glasses and fixed using chilled (−20 °C) methanol. Fixed keratinocytes were treated with 5% goat serum for 1 h at room temperature to block the nonspecific sites. The keratinocytes were incubated overnight with primary antibodies (1:100 dilution factor) at 4 °C, followed by three washes with PBS. To detect the AE2 antibody, keratinocytes were treated with goat immunoglobulin G (IgG)-tagged with rhodamine (1:50 dilution factor, Jackson ImmunoResearch, anti-rabbit: 111-025-144) for 1 h at room temperature. Following incubation, keratinocytes were washed thrice with PBS, and the cover glasses were mounted onto glass slides using Fluoromount-G™ with 4,6-diamidino-2-phenylindole (DAPI, 17984-24, Electron Microscopy Sciences, Hatfield, PA, USA) and incubated overnight at 4 °C. The slides were analyzed using an LSM 700 Zeiss confocal microscope (Carl Zeiss, Oberkochen, Germany) and the ZEN software (Carl Zeiss, Oberkochen, Germany).

### 4.8. Quantitative Real-Time Polymerase Chain Reaction (qRT-PCR) and Reverse Transcription-Polymerase Chain Reaction (RT-PCR)

Total RNA was extracted from cardiac tissues using the Hybrid-Ribo^Ex^ extraction system (Gentaur, Kampenhout, Belgium), according to the manufacturer’s instructions. RNA was quantified using the Spectrophotometer ND-1000 (Thermo Fisher Scientific) and was amplified according to the manufacturer’s protocol, using the TOPscript™ RT-PCR kit from Enzynomics (Daejeon, Korea). The human primers used are listed in Table 1. Quantitative RT-PCR was performed by CFX384 Touch Real-Time PCR Detection System (BioRad, Hercules, CA, USA) with Thunderbird™ SYBR qPCR Mix (QPK-201, TOYOBO, Osaka, Japan), according to manufacturer’s protocol. The C_T_ values were determined with the default threshold setting. Relative expressions for the target RNAs were determined by the comparative C_T_ (2^−ΔΔCt^) method applied for data analysis. The qRT-PCR cycling protocol used was as follows: denaturation at 95 °C for 1 min, followed by 45 cycles at 95 °C for 15 s, an annealing step for 45 s, an extension step at 95 °C for 15 s and 60 °C for 15 s. A final extension was carried out at 95 °C for 15 s. The RT-PCR cycling protocol used was as follows: denaturation at 95 °C for 5 min, followed by 45 cycles at 95 °C for 30 s, an annealing step for 1 min (Table 1 and Table 2), and an extension step at 72 °C for 1 min. A final extension was carried out at 72 °C for 10 min. PCR products were electrophoresed on 1% agarose gels. Bands were visualized and acquired using a CCD camera and were scanned using the GelDoc^XR^ imaging system (Bio-Rad, Hercules, CA, USA). 

### 4.9. DNA Transfection

Plasmid DNA transfection was performed by jetPRIME as per manufacturer’s protocol (Polyplus-transfection, Alsace, France). Plasmid DNA was diluted in 200 μL of jetPRIME buffer and 4 μL of jetPRIME reagent, and was incubated for 10 min at RT. Add transfection mix to the cells in serum containing medium and was incubated at 37 °C in a humidified incubator with 5% CO_2_ and 95% air. After further 4 h incubation, the medium was replaced with fresh DMEM containing FBS. The cells were cultured and used for experiment after 48 h of transfection.

### 4.10. Statistical Analyses

All data from the indicated number of experiments were expressed as the mean ± standard error of the mean (SEM). The statistical differences between mean values obtained from the two or more sample groups were analyzed using paired Student’s *t*-test. Two independent sample datasets come from distributions with different of two different groups. Statistical significance was determined by analysis of variance for each experiment (* *p* < 0.05, ** *p* < 0.01, *** *p* < 0.001).

## Figures and Tables

**Figure 1 ijms-21-08429-f001:**
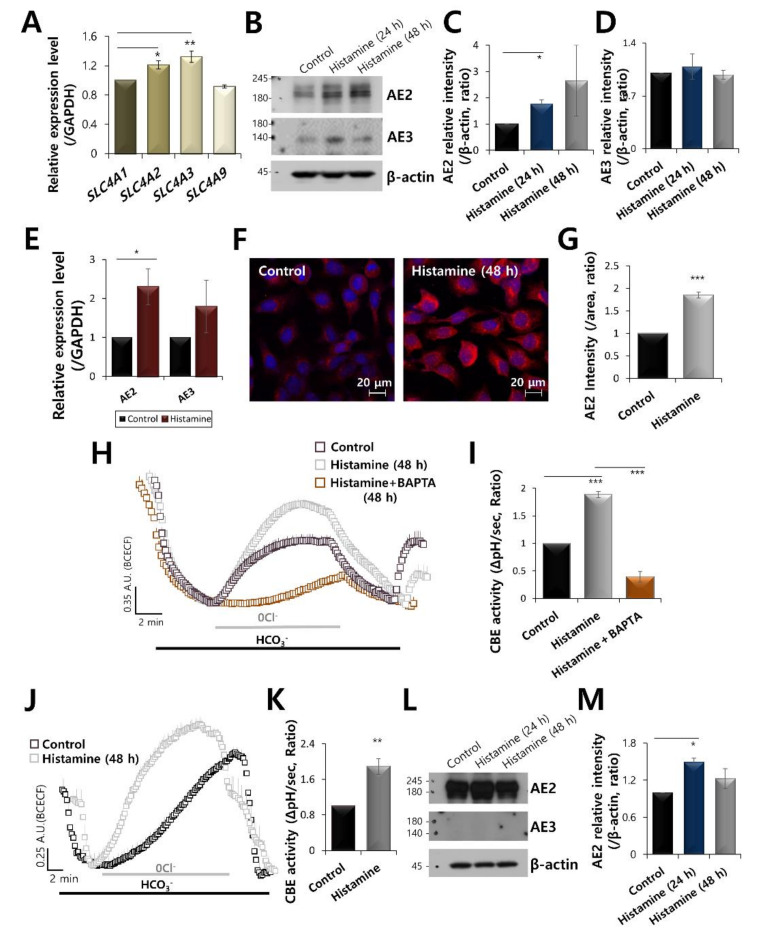
AE2 is activated by the stimulation of histamine in keratinocytes. (**A**) mRNA expression of SLC4 family receptors (*SLC4A1, A2, A3,* and *A9*) in HaCaT keratinocytes. (**B**) The protein expression of AE2, AE3, and β-actin during histamine treatment at 24 and 48 h in HaCaT keratinocytes. The β-actin was used as a loading control. Analysis of AE2 expression (**C**) and AE3 expression (**D**) with or without histamine in HaCaT cells. The bars indicate the mean ± SEM of data (* *p* < 0.05, *n* = 4). (**E**) mRNA expression of AE2 and AE3 with or without histamine stimulation in HaCaT cells (* *p* < 0.05). (**F**) Immunostaining of AE2 (red) and nucleus (DAPI, blue) in the presence of 500 nM histamine at 48 h. (**G**) The bars indicate the mean ± SEM of the AE2 membrane intensity determined from three experimental replicates (*** *p* < 0.001, *n* = 3). (**H**) CBE activity of HaCaT keratinocytes with (grey open square) and without (control, black open square) 500 nM histamine and with co-stimulation of 500 nM histamine and 10 μM BAPTA-AM (orange open square) at 48 h. Averaged traces were represented. (**I**) Analysis of CBE activity. The bars indicate the means ± SEM of the number of experiments (*** *p* < 0.001, *n* = 4). (**J**) CBE activity of primary keratinocytes with (grey open square) and without (control, black open square) 500 nM histamine at 48 h. (**K**) Analysis of CBE activity of primary epidermal keratinocytes. The bars indicate the means ± SEM of the number of experiments (** *p* < 0.01, *n* = 3). (**L**) The protein expression of AE2, AE3, and β-actin during histamine treatment at 24 and 48 h in primary epidermal keratinocytes. The β-actin was used as a loading control. (**M**) Analysis of AE2 expression with or without histamine stimulation in primary epidermal keratinocytes. The bars indicate the mean ± SEM of data (* *p* < 0.05, *n* = 4).

**Figure 2 ijms-21-08429-f002:**
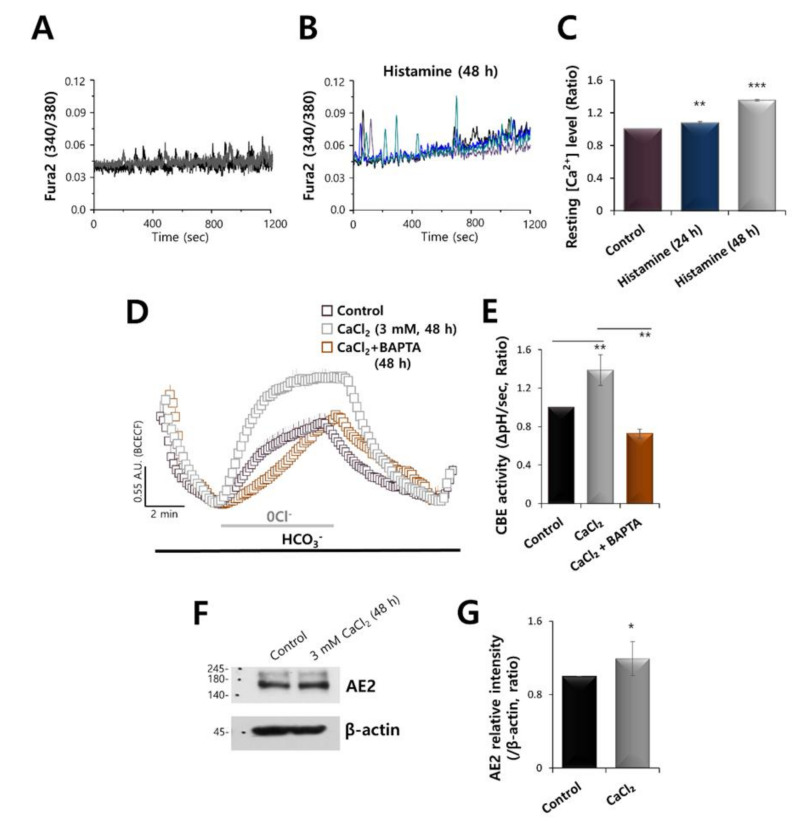
Increased calcium by histamine stimulation enhanced CBE activity and AE2 expression in HaCaT keratinocytes. Stimulation with 500 nM histamine induced an increase in Ca^2+^ in HaCaT keratinocytes for the control (**A**) and at 48 h (**B**). (**C**) Analysis of the resting Ca^2+^ level. Bars indicate the mean ± SEM of the number of experiments (** *p* < 0.05, *** *p* < 0.001, *n* = 20). (**D**) CBE activity of HaCaT keratinocytes with (gray open square) and without (control, black open square) 3 mM CaCl_2_ and with co-stimulation of 3 mM CaCl_2_ and 10 μM BAPTA-AM (orange open square) at 48 h. (**E**) Analysis of CBE activity. Bars indicate the mean ± SEM of the number of experiments. (** *p* < 0.01, *n* = 4). (**F**) Protein expression of AE2 and β-actin with 3 mM CaCl_2_ at 48 h. The β-actin was used as a loading control. (**G**) Analysis of AE2 expression with or without 3 mM CaCl_2_ at 48 h. Bars indicate the means ± SEM (* *p* < 0.05, *n* = 4).

**Figure 3 ijms-21-08429-f003:**
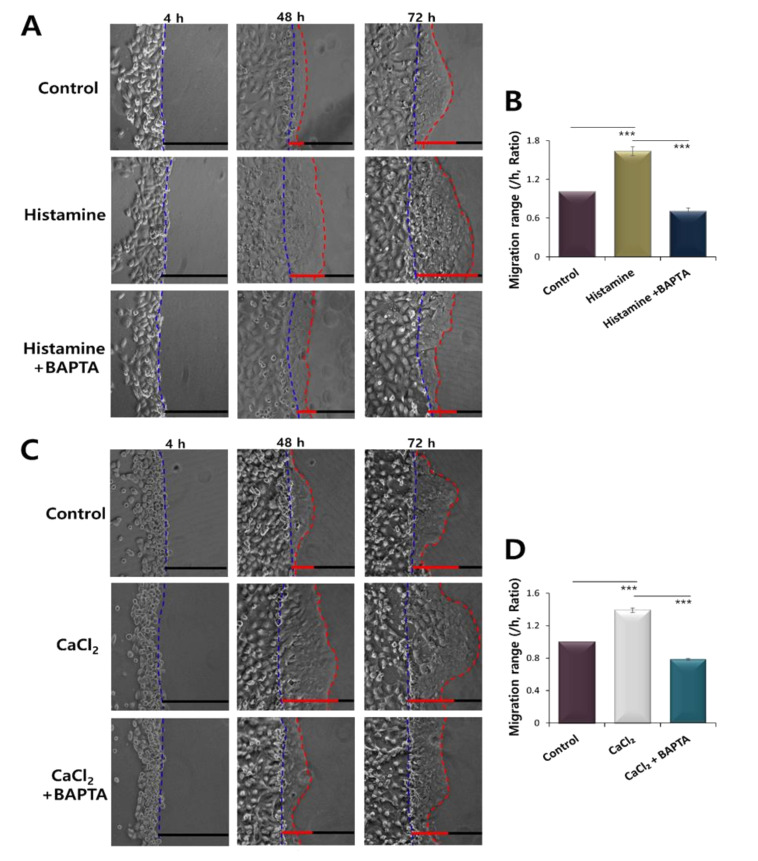
CaCl_2_ and histamine stimulation enhanced HaCaT cell migration. (**A**) Time dependent representative images of HaCaT keratinocytes migrating (4, 48, and 72 h) towards agarose spots in the absence or presence of 500 nM histamine in media with or without 10 μM BAPTA-AM. The direction of migration across the boundary of the agarose spot is shown as a dashed curve (blue dotted lines). The red dotted lines indicate the lineage of keratinocytes that moved into the spots. (**B**) Analysis of migration range per hour in agarose spots in the absence or presence of 500 nM His in the medium. Bars indicate the means ± SEM of the number of experiments (*** *p* < 0.001, *n* = 3). (**C**) Time dependent representative images of HaCaT keratinocytes migrating (4, 48, and 72 h) towards agarose spots in the absence or presence of 3 mM CaCl_2_ in media in media with or without 10 μM BAPTA-AM. (**D**) Analysis of migration range per hour in agarose spots in the absence or presence of 3 mM CaCl_2_ in the media with or without 10 μM BAPTA-AM. Bars indicate the means ± SEM of the number of experiments (*** *p* < 0.001, *n* = 3).

**Figure 4 ijms-21-08429-f004:**
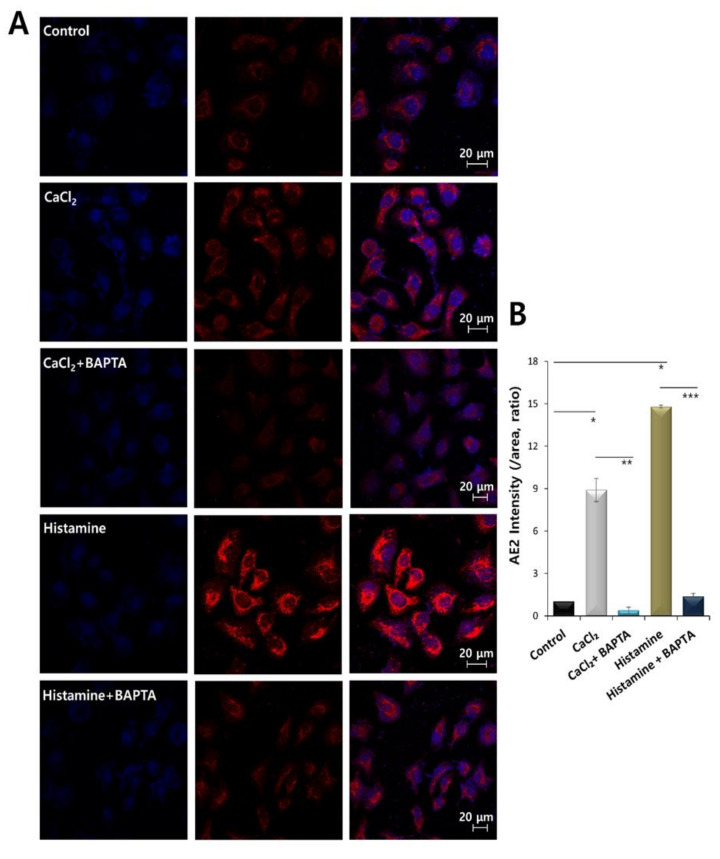
AE2 expression was dependent on histamine and CaCl_2_ stimulation. (**A**) Immunostaining of AE2 (red) and nucleus (DAPI, blue) in the presence of 500 nM histamine or 3 mM CaCl_2_ with or without 10 μM BAPTA-AM at 48 h. (**B**) The bars indicate the mean ± SEM of the AE2 membrane intensity determined from three experimental replicates (* *p* < 0.05, ** *p* < 0.01, *** *p* < 0.001, *n* = 4).

**Figure 5 ijms-21-08429-f005:**
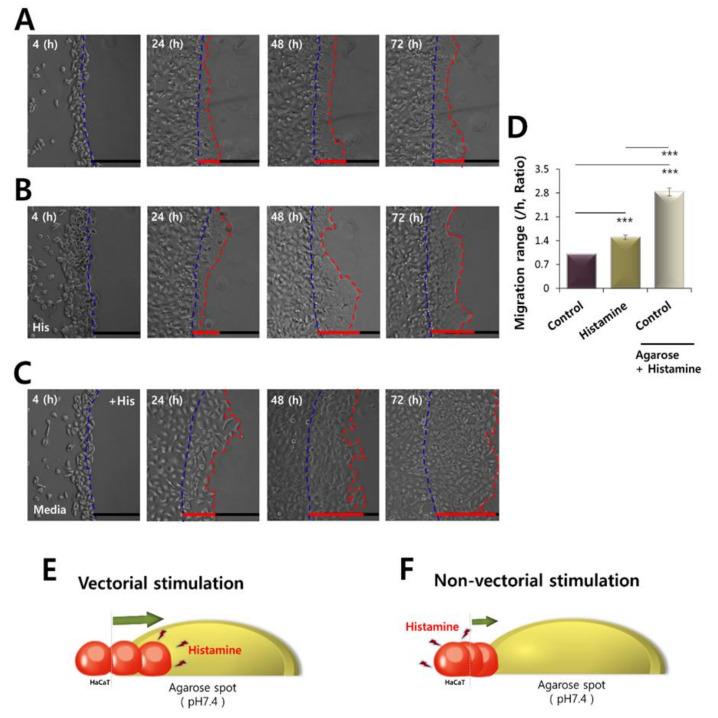
Motility of HaCaT cells was driven more by vectorial chemotaxis. Time dependent representative images of HaCaT keratinocytes migrating (4, 24, 48, and 72 h) towards agarose spots (**A**) in PBS (pH7.4)-containing agarose spots, (**B**) in the presence of 500 nM His-containing media and in PBS (pH7.4)-containing agarose spots, and (**C**) in the 500 nM His-containing agarose spots. The direction of migration across the boundary of the agarose spot is shown as a dashed curve (blue dotted lines). The red dotted lines indicate the lineage of keratinocytes that moved into the spots. (**D**) Analysis of migration range per hour in agarose spots in the absence or presence of 500 nM His. Bars indicate the means ± SEM of the number of experiments (*** *p* < 0.001, *n* = 4). (**E**) Schematic illustration of the vectorial stimulation toward the histamine-containing agarose spot. (**F**) Schematic illustration of non-vectorial stimulation of histamine-containing media toward the agarose spot.

**Figure 6 ijms-21-08429-f006:**
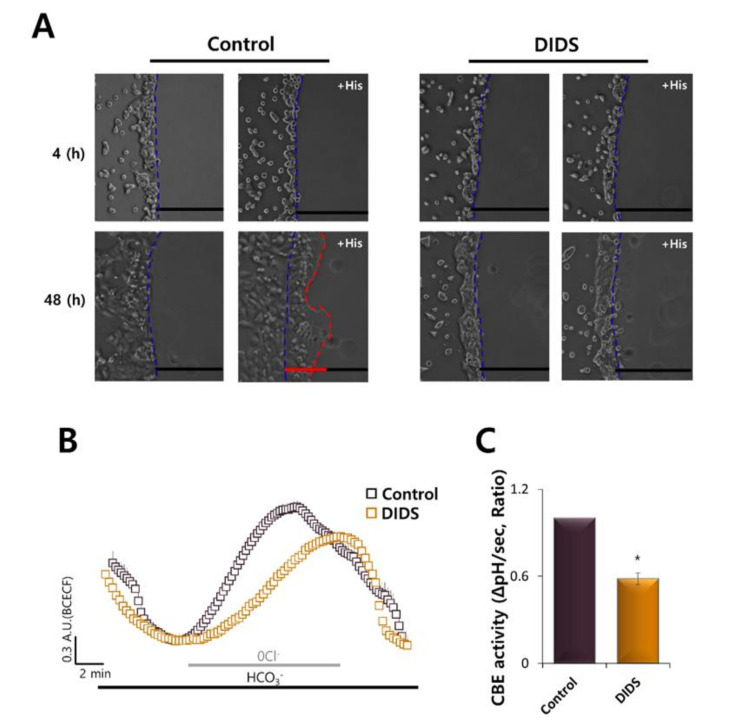
Inhibition of transporters by DIDS attenuated HaCaT migration. (**A**) Time-dependent representative images of HaCaT keratinocytes migrating (4 and 48 h) towards agarose spots containing PBS (pH 7.4) with and without 500 nM His and in the absence or presence of 500 μM DIDS-containing media. The direction of migration across the boundary of the agarose spot is shown as a dashed curve (blue dotted lines). The red dotted lines indicate the lineage of keratinocytes that moved into the spots. (**B**) CBE activity of HaCaT keratinocytes with 500 μM DIDS (orange open square) and without (control, black open square) at 48 h. Averaged traces were represented. (**C**) Analysis of CBE activity. Bars indicate the means ± SEM of the number of experiments (* *p* < 0.05, *n* = 3).

**Figure 7 ijms-21-08429-f007:**
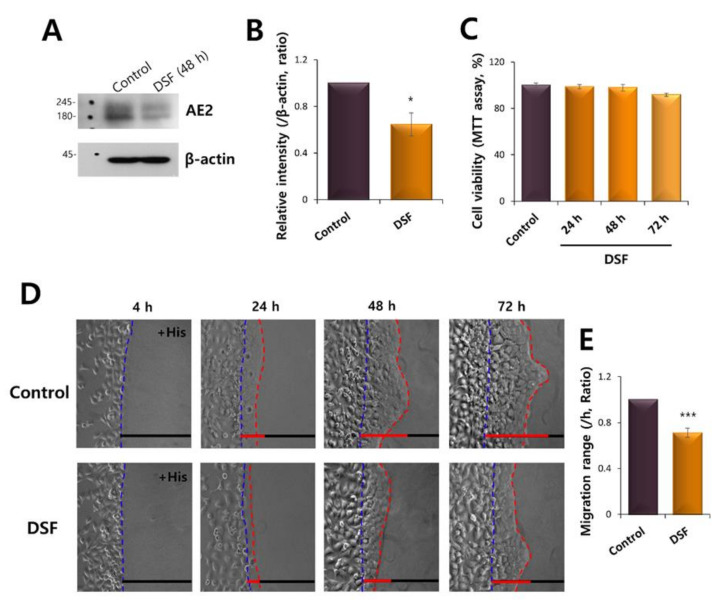
AE2 inhibition attenuated the vectorial movement of HaCaT cells. (**A**) The protein expression of AE2 and β-actin during 2 μM disulfiram (DSF) treatment at 48 h in HaCaT keratinocytes. The β-actin was used as a loading control. (**B**) Analysis of AE2 intensity in the presence of DSF at 48 h. (**C**) The cell viability in presence of 2 μM DSF at 24, 48, and 72 h. (**D**) Time-dependent representative images of migrated HaCaT cells at 4, 24, 48, and 72 h, towards agarose spots containing PBS (pH 7.4) with 500 nM His, with or without 2 μM DSF in the media. The direction of migration across the boundary of the agarose spot is shown as a dashed curve (blue dotted lines). The red dotted lines indicate the lineage of keratinocytes that moved into the spots. (**E**) Analysis of migration range compared to the control in the presence of 2 μM DSF in the media. Bars indicate the means ± SEM (*** *p* < 0.001, *n* = 5).

**Figure 8 ijms-21-08429-f008:**
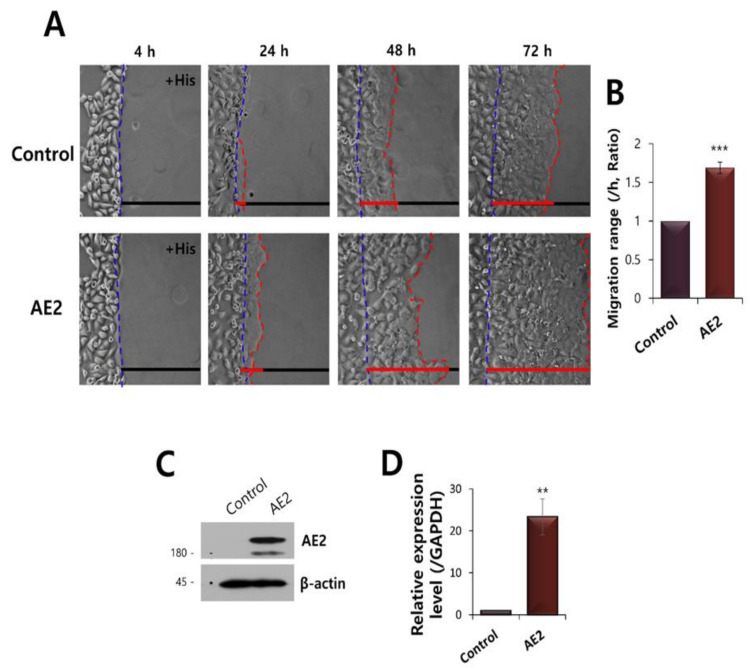
Overexpressed AE2 enhanced vectorial HaCaT migration. (**A**) Time-dependent representative images of HaCaT keratinocytes migrating towards agarose spots containing PBS (pH 7.4) with 500 nM His in control or AE2-overexpressed HaCaT cells. The direction of migration across the boundary of the agarose spot is shown as a dashed curve (blue dotted lines). The red dotted lines indicate the lineage of keratinocytes that moved into the spots. (**B**) Analysis of migration range of AE2-overexpressed HaCaT cells compared to the control. Bars indicate the means ± SEM (*** *p* < 0.001, *n* = 4). (**C**) Protein expression of AE2 in AE2-overexpressed HaCaT cells. The β-actin was used as a loading control. (**D**) mRNA expression of AE2 in AE2-overexpressed HaCaT cells (** *p* < 0.01, *n* = 3).

**Figure 9 ijms-21-08429-f009:**
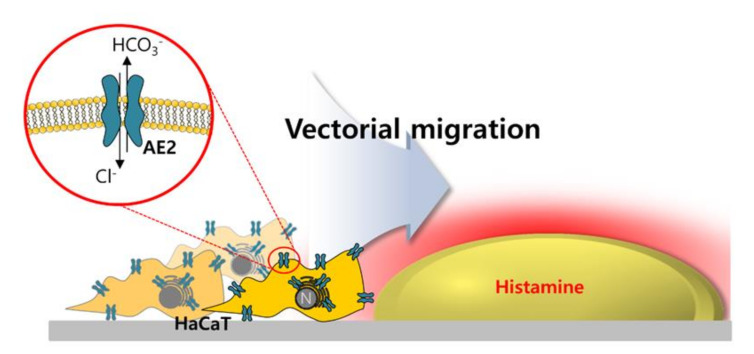
Schematic illustration of vectorial HaCaT migration through the involvement of AE2.

**Table 1 ijms-21-08429-t001:** Human quantitative real-time polymerase chain reaction (qRT-PCR) primer sequences

Genes.	Tm (°C)	Sequences (5′ → 3′)
*SLC4A1*	60	(F) CAC ACA ACT TCA GGC CCC TC
		(R) AGA GCC TGC TGT CTC CTA CC
*SLC4A2*	59	(F) AGT TGG GAG AAG TTG GGA GC
		(R) CAT AAC CCG CTC GCT CTG G
*SLC4A3*	58	(F) GTT TGG GGA CCT CAT CAG CA
		(R) ATG TGT GCC GGT GAA ACT CA
*SLC4A9*	58	(F) GAC CCC AGG AAA CAG CAT GA
		(R) CAC CCT CAG GTC AGG AGG TA
*GAPDH*	59	(F) CCG CAT CTT CTT TTG CGT CG
		(R) TTC CCG TTC TCA GCC TTG AC

**Table 2 ijms-21-08429-t002:** Human reverse transcription-polymerase chain reaction (RT-PCR) primer sequences

Genes	Tm (°C)	Sequences (5′ → 3′)
Histamine 1 receptor (R)	58	(F) GAC TGT GTA GCC GTC AAC CGG A
		(R) TGA AGG CTG CCA TGA TAA AAC C
Histamine 2R	56	(F) TCG TGT CCT TGG CTA TCA C
		(R) CTT TGC TGG TCT CGT TCC T
Histamine 3R	70	(F) TCA GCT ACG ACC GCT TCC TGT CGG TCA C
		(R) TTG AGT GAG CGC GGC CTC TCA GTG CCC C
Histamine 4R	63	(F) GAA TTG TCT GGC TGG ATT AAT TTG CTA ATT TG
		(R) AAG AAT GAT GTG GTG ATG GCA AGG ATG TAC C
GAPDH	62	(F) CAT GGC ACC GTC AAG GCT GAG
		(R) CTT GGC CAG GGG TGC TAA GC

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
