# Peer review of "Intracellular Ca2+-Mediated AE2 Is Involved in the Vectorial Movement of HaCaT Keratinocyte"

_ijms, 2020, doi:10.3390/ijms21228429_

Round 1
Reviewer 1 Report
The manuscript by S Hwang et al was considerably improved. Non-conclusive data have been deleted and new convincing experiments added. The reported data strongly suggest that a AE2-mediated intracellular calcium concentration increased is involved in HaCaT cell migratory capability.
However, some major new issues appeared.
- From the new data about qRT-PCR expression analysis, it can be concluded that AE1 and AE4 are expressed in HaCaT cells, with a slightly lower level than AE2 and AE3 but a similar expression level than GAPDH (deltaCt values were not presented). Therefore they may be involved in the described histamine effect. Please comment this point in the discussion.
- The Authors suggested that AE2 overexpression enhanced the vectorial HaCaT cell migration (paragraph 2.8 and Fig 8). However, there is no proof that AE2 is actually overexpressed. Western blotting or qRT-PCR data are mandatory.
Minor points:
- Fig 1E legend, the CBE values corresponding to HaCaT cells in the presence of histamine are represented by grey squares not pink ones as stated.
- Fig 1B, it is surprising to see an AE3-antibody reactive band at 60 kDa. Please explain. By the way, the anti-AE3 antibody is not described in the Material section.
- Fig 1I and line 71, it is not clear for this reviewer that AE2 expression was enhanced. Please quantify.
- Fig 2F, the expression of AE2 did not appear to be enhanced by CaCl2 treatment as stated in the text (line 102). Please quantify.
- Lines 169-170, please delete the sentence “Cellular migration or invasion … exchangers [19]” because it is confusing and should be rather placed in the Introduction section.
- Fig S1 is not commented.
Author Response
Dear reviewer and editor,
The manuscript has been edited to make appropriate information and additional data to this body of work.
Responses to comments of reviewer as below:
The manuscript by S Hwang et al was considerably improved. Non-conclusive data have been deleted and new convincing experiments added. The reported data strongly suggest that a AE2-mediated intracellular calcium concentration increased is involved in HaCaT cell migratory capability.
However, some major new issues appeared.
- From the new data about qRT-PCR expression analysis, it can be concluded that AE1 and AE4 are expressed in HaCaT cells, with a slightly lower level than AE2 and AE3 but a similar expression level than GAPDH (deltaCt values were not presented). Therefore they may be involved in the described histamine effect. Please comment this point in the discussion.
- Response: We appreciate your valuable comment and added qRT-PCR data in Figure 1. The expression of AE3 was statistically no difference with histamine stimulation. Thus we focused on the AE2 and we added this point in Result and Discussion section.
- The Authors suggested that AE2 overexpression enhanced the vectorial HaCaT cell migration (paragraph 2.8 and Fig 8). However, there is no proof that AE2 is actually overexpressed. Western blotting or qRT-PCR data are mandatory.
- Response: We appreciate your valuable comment and added data for proof the expression of AE2 with Western blotting and qRT-PCR techniques in Fig 8C and 8D.
Minor points:
- Fig 1E legend, the CBE values corresponding to HaCaT cells in the presence of histamine are represented by grey squares not pink ones as stated.
- Response: We appreciate your valuable comment and made it correctly.
- Fig 1B, it is surprising to see an AE3-antibody reactive band at 60 kDa. Please explain. By the way, the anti-AE3 antibody is not described in the Material section.
- Response: We appreciate your valuable comment and apologize the confusion and made the AE3 band correctly. And antibody information was added in Material section.
- Fig 1I and line 71, it is not clear for this reviewer that AE2 expression was enhanced. Please quantify.
- Response: We appreciate your valuable comment and added quantified data in Fig 1M.
- Fig 2F, the expression of AE2 did not appear to be enhanced by CaCl2 treatment as stated in the text (line 102). Please quantify.
- Response: We appreciate your valuable comment and added quantified data in Fig 2G.
- Lines 169-170, please delete the sentence “Cellular migration or invasion … exchangers [19]” because it is confusing and should be rather placed in the Introduction section.
- Response: We appreciate your valuable comment and removed the sentence and rearranged it in introduction section.
- Fig S1 is not commented.
- Response: We labelled the supplementary figure S1 correctly and commented in supplementary materials section end of manuscript as following the journal format.

Reviewer 2 Report
Authors have responded to my comments
Author Response
We appreciate your favorable consideration.
Round 2
Reviewer 1 Report
Although the Authors did not respond to my first point and did not comment about a possible role of AE1 and AE4, they satisfactorily answered to all the others and improved their manuscript.
This manuscript is a resubmission of an earlier submission. The following is a list of the peer review reports and author responses from that submission.
Round 1
Reviewer 1 Report
In this manuscript, S Hwang et al report about their data concerning the involvement of anion exchanger AE2 in the migration of keratinocytes. They demonstrated using an in vitro migration assay and HaCaT keratinocyte cell line that histamine stimulated, in a chemotaxis way, the cell migration through increasing [Ca++]i and AE2 expression and activity. This is a very original and promising study, with clear data. However, some major issues need to be addressed.
- My main point concerns the HaCaT cells. Indeed, it is a transformed cell line that is known to be very different from keratinocytes in terms of metabolism, protein expression, differentiation, etc. Therefore I wonder what the real in vivo significance of the obtained data is. Major results have to be confirmed with primary keratinocytes. A wound healing model using in vitro assay or in vivo animal model would give more confidence in the results. At minimum, HaCaT should be mentioned in the title and in the abstract, and the significance of the results discussed.
- The histamine receptor antagonist clemastine inhibited cell migration, as shown on Fig 3. Did the migration value return to the control (without histamine) one? What is the effect of an HR agonist?
- Fig 6, a control with a non-relevant antibody is necessary. The best would be to use an anti-AE3 antibody.
- Data reported in paragraph 2.7 must be confirmed by quantitative RT-PCR. In 2020, semi-quantitative PCR is not allowed to compare the expression of genes. Please note that KRT1 and KRT10 are not proliferative markers since they are not expressed in basal keratinocytes, it is KRT5 and KRT14.
Minor points:
- Fig 1, please explain why you did not test the expression of AE4? The staining of cells is not membranous as stated in the legend of part (D). Be careful about the colors in part (E), pink or grey (idem in the other figures). It would be interesting to note that AE2 is highly immunodetected in the basal and first suprabasal keratinocyte layer of the human epidermis (ProteinAtlas website data).
- Please introduce clemastine fumarate on line 98. Please use the term “histamine receptor antagonist” rather than “antihistamine agent”.
- Lines 99-120. The sentences appear contradictory: “clemastine attenuated HaCaT migration” vs “clemastine fumarate did not modulate cellular migration”.
- Fig 2G, the AE2 antibody detects 3 bands (vs one in Fig 1B); different isoforms? Please comment.
- Fig4D, were the significances calculated against the control (without histamine)? What about histamine vs control+histamine?
- Lines 147-148, please delete the sentence that is rather confusing; indeed from the beginning your aim is to test the involvement of ion exchangers.
- A schematic conclusion would be informative.
Reviewer 2 Report
Major
- There is a disjoint between the short term- seconds and minutes- to the long term effects. The changes of calcium and effect of external calcium and bicarbonate are shown on the seconds time scale, but the expression effects are shown at the days time scale. There is nothing to join them and it is unclear if the described calcium changes are what mediate the expression effects at day 1 and 2
- The data shown in figure 3a with clemastine fumarate is not consistent with the conclusions summarized in figure 3c. The raw data does not support the summary.
- The magnification and scale in figure 5 is different than the rest of the figure. The scale should be consistent and indicated.
- I am having a hard time understanding the blue line and red line and the vectorial vs non vectorial movement. I assume the blue is where the cells were seeded and red is how far they migrated. How can everything to the right of the blue have histamine and still have the cells continue to migrate past that? it would seem they would stop since the histamine concentration is no longer changing or they would start moving randomly since again there is not histamine gradient to the right of the blue.
- Why would anti-AE2 block the movement of keratinocytes? It is very rare that antibodies are blocking antibodies- meaning they block function in vivo. Further this is a “fast track” antibody- meaning it has not been fully tested. Third, this is made against epitopes in a.a. 200-300. These are intracellular. How can the antibody get inside and cause block. That seems to be virtually impossible.
- The data in figure 8 seem to argue that AE2 expression is not what is critical to migration since histamine + anti-histamine still cause the increase in AE2. It is at odds with data in figure 3D.
Minor:
- Probability is not an exact definable quantity. It cannot be equal. So p= something in figure 1, 2, 3, etc…is not correct. Rather it is p> or p< than something as figure 2 legend
- Why is the clemastine fumarate levels reported as weight per volume? What is the concentration. That should be the case for all agents
Reviewer 3 Report
Review: MS. Intracellular Ca2+ mediated AE2 is involved in the vectorial movement of keratinocytes by Hwang and colleagues.
This is a well thought of and properly conducted experimental study examining the background and mechanisms governing keratinocyte migration in wound healing. The data supports that it is the Ca2+ exerted stimulation through the involvement of the AE2 machinery that facilitates the vectorial migration.
The authors have in an elegant manner and through several experimental steps provided good evidence supporting their hypothesis.
The manuscript would merit by a short description of the more general from a wound healing perspective and clinical consequences that these findings may lead to.
Reviewer 4 Report
Major concerns that need to be addressed.
Histamine enhanced HaCaT cell migration, however the H1R antihistamine agent clemestine did not block migration or Calcium levels. Did the authors assess if clementine blocks histamine-induced calcium release? This is an important experiment to show that histamine is mediating migration t via calcium release (Figures 3 and Supplem. Figure 3).
In Figure 7A, clemestine seems to enhance histamine-induced AE2 expression. Was this quantified? This contradicts their data in Figure 1 showing that histamine mediates AE2 expression. Is there a feedback mechanism for histamine signaling and AE2 expression? This is not discussed in the manuscript. Moreover, Figure 7C indicates that histamine-induced CBE activity is not mitigated by clemestine, which also contradicts their conclusion.
In Figure 7B, clemestine enhaces MMP9 expression compared to control, however it mitigates histamine-induced MMP9 expression. How do the authors justify this?
Very poor discussion. The data generated was not appropriately discussed for the aspects mentioned above, as well as for most of the data in this manuscript.
Minor:
Provide quantification of migration range shown in Figure 5.